# BRAC+: Going Deeper with Behavior Regularized Offline Reinforcement Learning

## Abstract

Online interactions with the environment to collect data samples for training a Reinforcement Learning agent is not always feasible due to economic and safety concerns. The goal of Offline Reinforcement Learning (RL) is to address this problem by learning effective policies using previously collected datasets. Standard off-policy RL algorithms are prone to overestimations of the values of out-of-distribution (less explored) actions and are hence unsuitable for Offline RL. Behavior regularization, which constraints the learned policy within the support set of the dataset, has been proposed to tackle the limitations of standard off-policy algorithms. In this paper, we improve the behavior regularized offline reinforcement learning and propose *BRAC+*. We use an analytical upper bound on KL divergence as the behavior regularizor to reduce variance associated with sample based estimations. Additionally, we employ state-dependent Lagrange multipliers for the regularization term to avoid distributing KL divergence penalty across all states of the sampled batch. The proposed Lagrange multipliers allow more freedom of deviation to high probability (more explored) states leading to better rewards while simultaneously restricting low probability (less explored) states to prevent out-of-distribution actions. To prevent catastrophic performance degradation due to rare out-of-distribution actions, we add a gradient penalty term to the policy evaluation objective to penalize the gradient of the Q value w.r.t the out-of-distribution actions. By doing so, the Q values evaluated at the out-of-distribution actions are bounded. On challenging offline RL benchmarks, BRAC+ outperforms the state-of-the-art model-free and model-based approaches.

## 1 Introduction

Reinforcement Learning (RL) has shown great success in a wide range of applications including board games (Silver et al., 2016), strategy games (Vinyals et al., 2019), energy systems (Zhang et al., 2019), robotics (Lin, 1992), recommendation systems (Choi et al., 2018), etc. The success of RL relies heavily on extensive online interactions with the environment for exploration. However, this is not always feasible in the real world as it can be expensive or dangerous (Levine et al., 2020).

Offline RL, also known as batch RL, avoids online interactions with the environment by learning from a static dataset that is collected in an offline manner (Levine et al., 2020). While standard off-policy RL algorithms (Mnih et al., 2013; Lillicrap et al., 2016; Haarnoja et al., 2018a) can, in theory, be employed to learn from an offline data, in practice, they perform poorly due to distributional shift between the behavior policy (probability distribution of actions conditioned on states as observed in the dataset) of the collected dataset and the learned policy (Levine et al., 2020). The distributional shift manifests itself in form of overestimation of the out-of-distribution (OOD) actions leading to erroneous Bellman backups.

Prior works tackle this problem via behavior regularization (Fujimoto et al., 2018b; Kumar et al., 2019; Wu et al., 2019; Siegel et al., 2020). This ensures that the learned policy stays "close" to the behavior policy. This is achieved by adding a regularization term that calculates the $f$-divergence between the learned policy and the behavior policy. Kernel Maximum Mean Discrepancy (MMD) (Gretton et al., 2007), Wasserstein distance and KL divergence are widely used (Wu et al., 2019). The regularization term is either fixed (Wu et al., 2019), or tuned via dual gradient descent (Kumar et al., 2019), or applied using a trust region objective (Siegel et al., 2020).

In this paper, we propose improvements to the Behavior Regularized Actor Critic (BRAC) algorithm presented in (Wu et al., 2019). To obtain the same, we observe that sample based estimation of divergence measures is computationally expensive and prone to higher variance. Therefore, to reduce variance, we derive an analytical upper bound on the KL divergence measure as the regularization term in the objective function. Moreover, we show that current works that apply the regularization term i.e. the divergence measure, on the entire batch, end up distributing the penalty over all states in the batch in amounts inversely proportional to the state's probability of occurrence in the batch. This needlessly restricts the deviation of highly explored states while allowing less explored ones to deviate farther leading to OOD actions. To address the same, we employ state dependent Lagrange multipliers for the regularization terms and automatically tune their strength using state-wise dual gradient descent. In addition, the performance of the learned agent trained using prior methods often deteriorates over the course of training. We found that if the learned Q function generalizes such that the gradient of the Q function w.r.t the OOD actions is monotonically increasing, behavior regularization fails to keep such actions within the support set. To mitigate this issue, we penalize the gradient of the Q function w.r.t the OOD actions by adding a gradient penalty term to the policy evaluation objective. This reduces the policy improvement at OOD actions to the problem of minimizing the divergence between the learned policy and the behavior policy.

We call our improved algorithm BRAC+ following (Wu et al., 2019). Our experiments suggest that BRAC+ outperforms existing state-of-the-art model-free and model-based offline RL algorithms in various datasets on the D4RL benchmark (Fu et al., 2020).

## 2 BACKGROUND

**Markov Decision Process** RL algorithms aim to solve Markov Decision Process (MDP) with unknown dynamics. A Markov decision process (Sutton & Barto, 2018) is defined as a tuple $< \mathcal{S}, \mathcal{A}, R, P, \mu >$, where $\mathcal{S}$ is the set of states, $\mathcal{A}$ is the set of actions, $R(s, a, s') : \mathcal{S} \times \mathcal{A} \times \mathcal{S} \to \mathbb{R}$ defines the intermediate reward when the agent transitions from state $s$ to $s'$ by taking action $a$, $P(s'|s,a) : \mathcal{S} \times \mathcal{A} \times \mathcal{S} \to [0,1]$ defines the probability when the agent transitions from state $s$ to $s'$ by taking action $a$, $\mu : \mathcal{S} \to [0,1]$ defines the starting state distribution. The objective of reinforcement learning is to select policy $\pi : \mu \to P(A)$ to maximize the following objective:

$$J(\pi) = \underset{\substack{s_0 \sim \mu, a_t \sim \pi(\cdot|s_t) \\ s_{t+1} \sim P(\cdot|s_t, a_t)}}{\mathbb{E}} [\sum_{t=0}^{\infty} \gamma^t R(s_t, a_t, s_{t+1})] \tag{1}$$

**Offline Reinforcement Learning** The goal of offline RL is to learn policy $\pi_\theta$ from a fixed dataset $\mathcal{D} = \{(s_i, a_i, s'_i, r_i)\}_{i=1}^N$ consisting of single step transitions $\{(s_i, a_i, s'_i, r_i)\}$. The dataset is assumed to be collected using a behavior policy $\pi_\beta$ which denotes the conditional distribution $p(a|s)$ observed in the dataset. Note that $\pi_\beta$ may consist of multi-modal policy distribution. In principle, standard off-policy RL algorithms using a replay buffer (Mnih et al., 2013; Lillicrap et al., 2016; Haarnoja et al., 2018a) can directly learn from $\mathcal{D}$. The key challenge resides in the policy evaluation step:

$$Q_\psi = \underset{\psi}{\arg\min}[(Q_\psi(s,a) - (r(s,a) + \gamma \mathbb{E}_{a' \sim \pi_\theta} Q_{\psi'}(s', a')))]^2 \quad \text{(policy evaluation)} \tag{2}$$

In this step, the target Q value depends on the learned policy. If the learned policy distribution $\pi_\theta$ diverges from the data distribution $\pi_\beta$, it results in evaluation of target Q values using out-of-distribution (OOD) actions. Such evaluations are prone to errors. Erroneous overestimation of values get exploited by the policy improvement, preventing the algorithm from learning useful policies. In order to avoid such cases, behavior regularization is adopted to force the learned policy to stay "close" to the behavior policy (Fujimoto et al., 2018b; Kumar et al., 2019; Wu et al., 2019).

## 3 IMPROVING BEHAVIOR REGULARIZED OFFLINE REINFORCEMENT LEARNING

In this section, we discuss and propose three non-trivial improvements to the Behavior Regularized Actor Critic (BRAC) offline reinforcement learning (Wu et al., 2019). BRAC (Wu et al., 2019)

augments either the policy improvement or the policy evaluation step with a penalty term to constrain the policy within the support set of the behavior policy. For simplicity of illustration, we only consider the policy improvement step (Wu et al., 2019):

$$\max_{\pi_\theta} \mathbb{E}_{(s,a,r,s')\sim\mathcal{D}}[\mathbb{E}_{a''\sim\pi_\theta(\cdot|s)}[Q_\psi(s,a'')] - \alpha\hat{D}(\pi_\theta(\cdot|s),\pi_\beta(\cdot|s))] \tag{3}$$

Here, $\hat{D}$ is a selected $f$-divergence (Csiszár, 1972) measure used for behavior regularization. Zooming into the regularization term, we get:

$$\sum_{s\in\mathcal{S}}(P_s\alpha)\hat{D}(\pi_\theta(\cdot|s),\pi_\beta(\cdot|s)) \tag{4}$$

In the equation above, $\mathcal{S}$ denotes the states that occur in a single sampled batch and $P_s$ is the probability of state $s$ in the sampled batch. The first two improvements that we propose are targeted towards the following two terms in Equation 4: (1) The divergence term $\hat{D}$ which for each fixed $\pi_\theta$ determines the impact actions have on the overall objective conditional on the state, and (2) the term $(P_s\alpha)$ which determines the impact the "divergence penalty" of each state will have on the overall objective function. Note that this term is dependent upon the probability of the occurrence of state in the sampled batch. For (1), we propose to use an analytical upper bound of KL divergence to reduce variance. For (2), we propose to decouple the impact of "divergence penalty" of each state from the probability of its occurrence. Detailed explanation of the two improvements and the rationale behind them follows in the next two sub-sections.

## 3.1 REGULARIZATION METHOD

Behavior regularization is used to constrain the learned policy with the support set of the behavior policy. In other words, it ensures that the "difference" between the probability distributions of the learned policy $\pi_\theta(\cdot|s)$ and behavior policy $\pi_\beta(\cdot|s)$ is small. The following divergence measures are most widely used in the community:

**Kernel MMD** Kernel Maximum Mean Discrepancy (MMD) (Gretton et al., 2007) was first introduced in (Kumar et al., 2019) to penalize the policy from diverging from the behavior policy:

$$\text{MMD}_k^2(\pi(\cdot|s),\pi_b(\cdot|s)) = \mathbb{E}_{x,x'\sim\pi(\cdot|s)}[K(x,x')] - 2\mathbb{E}_{\substack{x\sim\pi(\cdot|s)\\y\sim\pi_b(\cdot|s)}}[K(x,y)] + \mathbb{E}_{y,y'\sim\pi_b(\cdot|s)}[K(y,y')] \tag{5}$$

where $K$ is a kernel function. Symmetric kernel functions such as Laplacian and Gaussian kernels are typically used (Gretton et al., 2007).

**KL divergence** For two probability distributions $P$ and $Q$ on some probability space $\chi$, the KL-Divergence from $Q$ to $P$ is given as

$$\mathcal{D}_{\text{KL}}(P,Q) = \int_{x\sim\chi} P(x)\log\frac{P(x)}{Q(x)}dx \tag{6}$$

KL divergence is assymetric and both $\mathcal{D}_{\text{KL}}(\pi_\theta(\cdot|s),\pi_b(\cdot|s))$ and $\mathcal{D}_{\text{KL}}(\pi_b(\cdot|s),\pi_\theta(\cdot|s))$ are valid. However, $\mathcal{D}_{\text{KL}}(\pi_b(\cdot|s),\pi_\theta(\cdot|s))$ is not suitable because it requires $\pi_\theta(a|s) \neq 0$ when $\pi_b(a|s) \neq 0$, for each $a$. It may assign $\pi_\theta(a|s) > 0$ when $\pi_b(a|s) = 0$, producing out of distribution actions. Therefore, $\mathcal{D}_{\text{KL}}(\pi_\theta(\cdot|s),\pi_b(\cdot|s))$ is used for regularization.

### 3.1.1 ANALYTICAL KL DIVERGENCE UPPER BOUND

All the existing behavior regularized methods estimate the divergence via samples (Kumar et al., 2019; Wu et al., 2019). While in theory it produces an unbiased estimator, it requires a large number of samples to reduce the variance.

To stabilize the performance, the key idea is to have a low variance estimator for KL divergence. We obtain this by deriving an upper bound on the KL divergence between the learned policy $\pi_\theta$ and the behavior policy $\pi_\beta$ that can be computed analytically.

Assume we learn $\pi_\beta$ using a latent variable model (e.g. VAE (Kingma & Welling, 2014)) with latent variable $Z$. According to the evidence lower bound (ELBO), we obtain $\log \pi_\beta(a|s) \geq \mathbb{E}_{z \sim q(z)}[\log p(a|s, z)] - \mathcal{D}_{\text{KL}}(q(z)||p(z))$, where $q(z)$ is the approximated posterior distribution and $p(z)$ is the prior. Then, the KL divergence is bounded by:

$$
\begin{aligned}
\mathcal{D}_{\text{KL}}(\pi_\theta(\cdot|s)||\pi_\beta(\cdot|s)) &= \mathbb{E}_{a \sim \pi_\theta}[\log \pi_\theta(a|s)] - \mathbb{E}_{a \sim \pi_\theta}[\log \pi_\beta(a|s)] \\
&\leq \mathbb{E}_{a \sim \pi_\theta}[\log \pi_\theta(a|s)] - \mathbb{E}_{a \sim \pi_\theta}[\mathbb{E}_{z \sim q(z)}[\log p(a|s, z)] - \mathcal{D}_{\text{KL}}(q(z)||p(z))] \\
&= \mathbb{E}_{a \sim \pi_\theta, z \sim q(z)}[\mathcal{D}_{\text{KL}}(\pi_\theta(\cdot|s)||p(\cdot|s, z)) + \mathcal{D}_{\text{KL}}(q(z)||p(z))] \quad (7)
\end{aligned}
$$

Thus, if we choose $p(\cdot|s, z)$ to be a tractable distribution, Equation 7 can be computed analytically. Note that although $p(\cdot|s, z)$ has a special form (e.g. Gaussian), the latent variable model can in theory represent any probability distribution without sacrificing the expressiveness. In practice, we choose $p(\cdot|s, z)$ to be a Gaussian distribution.

## 3.2 State Dependent Lagrange Multiplier

From the term $(P_s \alpha)$ in Equation 4, one can infer that the "penalty budget" of divergence measure gets distributed among the states in amounts inversely proportional to the probability of the occurrence of the states in the sampled batch. If we assume the batches are sampled uniformly, this implies that the states which are more numerous in the offline dataset $\mathcal{D}$ i.e. the states that have been explored more thoroughly are restricted from deviating from the behavior policy. Whereas the less numerous state that have had limited exploration enjoy more freedom for deviation. This is undesirable as restricting highly explored states is overly conservative while allowing less explored states to deviate can lead to OOD actions.

To address this we add a state wise constraint for KL divergence as follows:

$$
\max_{\pi_\theta} \mathbb{E}_{s \sim \mathcal{D}}[\mathbb{E}_{a' \sim \pi_\theta(\cdot|s)}[Q_\psi(s, a')]] \text{ s.t. } \mathcal{D}_{KL}(\pi_\theta(\cdot|s), \pi_\beta(\cdot|s)) \leq \epsilon_{\text{KL}}, \forall s \quad \text{(policy improvement)}
$$

This translates to assignment of state dependent Lagrange multipliers when solved using dual gradient descent:

$$
\max_{\pi_\theta} \mathbb{E}_{s \sim \mathcal{D}}[\mathbb{E}_{a' \sim \pi_\theta(\cdot|s)}[Q_\psi(s, a') - \alpha(s) \mathcal{D}_{\text{KL}}(\pi_\theta(\cdot|s), \pi_\beta(\cdot|s))]] \quad (8)
$$

Maintaining a Lagrange multiplier for each state is impractical for large datasets. In practice, we parameterize $\alpha(s)$ with a neural network.

## 3.3 Gradient penalized policy evaluation

The fundamental challenge in offline reinforcement learning is to mitigate the impact of erroneous Q values that are evaluated at out-of-distribution actions and used in policy evaluation. Due to the limited representation capacity of neural networks, such actions are unavoidable for large datasets, even with state-dependent behavior regularization. CQL (Kumar et al., 2020) resolves this problem by optimizing a conservative lower bound of the Q value. **The key idea in this paper is to bound the Q value at the out-of-distribution actions such that their values are not greater than the Q value of in-distribution actions.** We achieve this by augmenting the policy evaluation step with a gradient penalty regularization term. To elaborate our approach, we first analyze the gradient of the policy improvement step:

$$
\nabla_\theta J \approx \frac{1}{N} \sum_{i=1}^{N} \nabla_{a_i} Q_\phi(s, a)|_{s=s_i, a=a_i} \nabla_\theta \pi_\theta(a|s)|_{s=s_i, a=a_i} - \alpha(s_i) \nabla_\theta \mathcal{D}_{\text{KL}}(\pi_\theta(\cdot|s), \pi_\beta(\cdot|s)) \quad (9)
$$

If the current policy produces out-of-distribution actions and the Q network erroneously generalizes in such a way that the gradient of Q network is monotonically increasing, this leads to the unbounded value of the Q network and the failure of the behavior regularization. We created a toy example to illustrate this phenomenon in Appendix A. The analysis suggests that if we penalize the gradient of the Q network with respect to the out-of-distribution actions such that they are close to zero,

---

**Algorithm 1** BRAC+: Improved Behavior Regularized Actor Critic

---

1: Train the behavior policy $\pi_\beta$ on the offline dataset $\mathcal{D} = \{(s_i, a_i, r_i, s_i')\}_{i=1}^N$ via maximum likelihood estimation: $\pi_\beta = \arg\max_{\pi_\beta} \sum_{i=1}^N \log \pi_\beta(a_i|s_i)$
2: Train initial policy: $\pi_\theta = \arg\min_{\pi_\theta \in \Pi} \mathcal{D}_{\mathrm{KL}}(\pi, \pi_\beta)$
3: Train initial Q network: $Q_\psi = \arg\min_\psi [(Q_\psi(s, a) - (r(s, a) + \gamma\mathbb{E}_{a' \sim \pi_\theta} Q_{\psi'}(s', a')))]^2$
4: **for** $e = 1 : E$ **do**
5:     **for** $t = 1 : T$ **do**
6:         Update Q network using Equation 10
7:         Update the policy using Equation 7
8:         Update $\alpha$ via dual gradient descent: $\alpha(s) \leftarrow \alpha(s) + \lambda_\alpha(\mathcal{D}_{\mathrm{KL}}(\pi_\theta(\cdot|s), \pi_\beta(\cdot|s)) - \epsilon_{\mathrm{KL}})$
9:         Update $\beta$ via dual gradient descent: $\beta \leftarrow \beta + \lambda_\beta(\mathcal{H}(\pi_\theta(\cdot|s)) - \mathcal{H}_0)$
10:        Update the target network $\psi' = \tau\psi + (1 - \tau)\psi'$
11:     **end for**
12: **end for**

---

performing policy improvement step simply reduces to minimizing the KL divergence. Inspired from (Gulrajani et al., 2017), we add a gradient penalty term to the policy evaluation step as:

$$\min_\psi \mathbb{E}_{(s,a)\sim\mathcal{D}}((Q_\psi(s, a) - r(s, a) + \gamma\mathbb{E}_{a'\sim\pi_\theta}Q_{\psi'}(s', a'))^2$$
$$+ \lambda\mathbb{E}_{a''\sim\pi_\theta(\cdot|s)}(||\nabla_{a''}Q_\psi(s, a'')||_2 f(\mathcal{D}_{KL}(\pi_\theta(\cdot|s), \pi_\beta(\cdot|s)) - \epsilon_{\mathrm{GP}})) \quad (10)$$

where $f$ is a non-decreasing function and $\epsilon_{GP}$ is the threshold for gradient penalty. In this paper, we set $f$ to be the `Heaviside` step function (or indicator). Other variants such as its soft version (`sigmoid` function) is left for future work. Setting $\epsilon_{GP} = \epsilon_{KL}$ is too conservative because it prevents generalization. In practice, we set $\epsilon_{GP}$ at epoch $t$ to be $\mu_{KL} + \sigma_{KL}$, where $\mu_{KL}$ and $\sigma_{KL}$ is the mean and standard deviation of the KL divergence for all the states in the dataset at epoch $t - 1$. In addition, we found that $\lambda = 0.1$ works well for all the tasks.

## 4 RELATED WORK

We briefly summarize prior works in offline RL and discuss their relationship with our approach. As discussed in Section 1, the fundamental challenge in learning from a static data is to avoid out-of-distribution actions (Levine et al., 2020). This requires solving two problems: 1) estimation of behavior policy, 2) quantification of out-of-distribution actions. We follow BCQ (Fujimoto et al., 2018b), BEAR (Kumar et al., 2019) and BRAC (Wu et al., 2019) by learning the behavior policy using a conditional VAE (Kingma & Welling, 2014). To avoid out-of-distribution actions, BCQ generates actions in the target values by perturbing the behavior policy. However, this is over-pessimistic in most of the cases. BRAC (Wu et al., 2019) constrains the policy using various sample-based $f$-divergence measures including MMD, Wasserstein distance and KL divergence with penalized policy improvement or policy evaluation. BEAR (Kumar et al., 2019) is an instance of BRAC with penalized policy improvement using MMD (Gretton et al., 2007). Sample-based estimation is computationally expensive and suffers from high variance. In contrast, our method uses an analytical upper-bound of the KL divergence to constrain the distance between the learned policy and the behavior policy. It is both computationally efficient and has low variance. (Siegel et al., 2020) solves trust-region objective instead of using penalty. CQL (Kumar et al., 2020) avoids estimating the behavior policy by learning a conservative Q function that lower-bounds its true value. Hyperparameter search is another challenging problem in offline RL. (Lee et al., 2020) uses a gradient-based optimization of the hyperparameter using held-out data. MOPO (Yu et al., 2020) follows MBPO Janner et al. (2019) with additional reward penalty on unreliable model-generated transitions. MBOP (Argenson & Dulac-Arnold, 2020) learns the dynamics mode, the behavior policy and a truncated value function to perform online planning. (Kidambi et al., 2020) learns a surrogate MDP using the dataset, such that taking out-of-distribution actions transit to the terminal state. The out-of-distribution actions are detected using the agreement of model ensembles.

# 5 EXPERIMENTS

Our experiments[1] aim to answer the following questions: 1) How does the performance of our improvements compare with state-of-the-art model-free and model-based offline RL methods? 2) How does the use of analytical variational upper bound on KL divergence for regularization term compare with sampled-based MMD? 3) How does the state-dependent Lagrange multiplier based regularization (state-wise regularizor) compare with global policy regularization? 4) Does the gradient penalized policy evaluation improve the stability during the training? To answer these questions, we evaluate our methods on a subset of the D4RL (Fu et al., 2020) benchmark. We consider three locomotion tasks (hopper, walker2d, and halfcheetah) and four types of datasets: 1) random (rand): collect the interactions of a run of random policy for 1M steps to create the dataset, 2) medium (med): collect the interactions of a run of medium quality policy for 1M steps as the dataset, 3) medium-expert (med-exp): run a medium quality policy and an expert quality policy for 1M steps, respectively, and combine their interactions to create the dataset, 4) mixed (medium-replay): train a policy using SAC (Haarnoja et al., 2018a) until the performance of the learned policy exceeds a pre-determined threshold, and take the replay buffer as the dataset. In addition, we consider more complex Adroit tasks (Rajeswaran et al., 2018) that requires controlling a 24-DoF robotic hand, using limited data from human demonstrations.

We compare against state-of-the-art model-free and model-based baselines, including behavior cloning, BEAR (Kumar et al., 2019) that constrains the learned policy within the support of the behavior policy using sampled MMD, BRAC-p/v (Wu et al., 2019) that constrains the learned policy within the support of the behavior policy using various sample-based $f$-divergences to penalize either the policy improvement (p) or the policy evaluation (v), CQL($\mathcal{H}$) (Kumar et al., 2020) that learns a Q function that lower-bounds its true value. We also compare against model-based approaches including MOPO (Yu et al., 2020) that follows MBPO (Janner et al., 2019) with additional reward penalties and MBOP (Argenson & Dulac-Arnold, 2020) that learns an offline model to perform online planning.

## 5.1 COMPARATIVE RESULTS

Table 1: Results for OpenAI gym (Brockman et al., 2016) environments in the D4RL (Fu et al., 2020) datasets. For each task, we train for 1 million gradient steps and report the performance by running the policy obtained at the last epoch of the training for 100 episodes, averaged over 4 random seeds with standard deviation. Each number is the normalized score as proposed in (Fu et al., 2020). Please refer to (Fu et al., 2020) for results on more baselines.

| Task Name | Model-Free | | | | Model-Based | |
|---|---|---|---|---|---|---|
| | BEAR | BRAC-p/v | CQL($\mathcal{H}$) | BRAC+ (Ours) | MOPO | MBOP |
| halfcheetah-rand | 25.1 | 24.1/31.2 | **35.4** | 26.4±1.0 | 31.9±2.8 | 6.3±4.0 |
| walker2d-rand | 7.3 | -0.2/1.9 | 7.0 | **16.7± 2.3** | 13.0±2.6 | 8.1± 5.5 |
| hopper-rand | 11.4 | 11.0/**12.2** | 10.8 | **12.5±0.3** | **13.3±1.6** | 10.8± 0.3 |
| halfcheetah-med | 41.7 | 43.8/**46.3** | 44.4 | **46.6±0.6** | 40.2± 2.7 | 44.6±0.8 |
| walker2d-med | 59.1 | 77.5/**81.1** | 79.2 | 75.1±3.5 | 14.0±10.1 | 41.0 ± 29.4 |
| hopper-med | 52.1 | 32.7/31.1 | **58.0** | 53.2±3.1 | 26.5±3.7 | 48.8 ± 26.8 |
| halfcheetah-med-exp | 53.4 | 44.2/41.9 | **62.4** | 61.2±2.8 | 57.9± 24.8 | **105.9± 17.8** |
| walker2d-med-exp | 40.1 | 76.9/81.6 | **98.7** | 95.3± 5.9 | 55.0± 19.1 | 70.2 ± 36.2 |
| hopper-med-exp | 96.3 | 1.9/0.8 | **111.0** | 112.9±0.1 | 51.7±42.9 | 55.1 ± 44.3 |
| halfcheetah-mixed | 38.6 | 45.4/**47.7** | 46.2 | 46.1±0.2 | **54.0±2.6** | 42.3 ± 0.9 |
| walker2d-mixed | 19.2 | -0.3/0.9 | 26.7 | **39.0±4.6** | **42.7±8.3** | 9.7 ± 5.3 |
| hopper-mixed | 33.7 | 0.6/0.6 | 48.6 | **72.7±18.9** | **92.5±6.3** | 12.4 ± 5.8 |

**Performance on multi-modal datasets** We first compare the performance on multi-modal datasets i.e. med-exp and mixed datasets. Results shown in Table 1 suggest that our method outperforms various model-free baselines on most of the multi-modal datasets, especially on hopper-mix

---

[1]Our code will be released at `https://github.com/xxx/xxx`.

Table 2: Results for Adroit tasks with human demonstrations in the D4RL (Fu et al., 2020) datasets. The numbers are reported by following the same procedure as in Table 1 except we run the policy obtained at the last epoch of training for 1000 episodes due to large variance across different runs.

| Task Name | BC | BEAR | BRAC-p/v | CQL($\mathcal{H}$) | CQL($\rho$) | BRAC+ (Ours) |
|---|---|---|---|---|---|---|
| pen-human | 34.4 | -1.0 | 8.1/0.6 | 37.5 | 55.8 | **64.9$\pm$ 1.6** |
| hammer-human | 1.5 | 0.3 | 0.3/0.2 | **4.4** | 2.1 | 3.9 $\pm$ 0.9 |
| door-human | 0.5 | -0.3 | -0.3/-0.3 | 9.9 | 9.1 | **11.5$\pm$ 1.2** |
| relocate-human | 0.0 | -0.3 | -0.3/-0.3 | 0.20 | **0.35** | 0.20 $\pm$ 0.11 |

and walker2d-mix by up to 1.5x. Compared with BEAR (Kumar et al., 2019), our performance improvement arises from the advantage of the KL divergence over the kernel MMD (see discussions in Appendix B). The choice of parameters in CQL($\mathcal{H}$) makes it too conservative to achieve higher performance.

**Performance on single-modal datasets**  The performance of our method on single-modal (rand and med) dataset outperforms or matches with baseline methods as evident from Table 1 except halfcheetah-random dataset. We observe that the performance is very sensitive to the choice of target policy entropy. We hypothesize that the our choice of target policy entropy in the halfcheetah-random task makes it hard to compose correct sub-optimal policies from a random collected dataset.

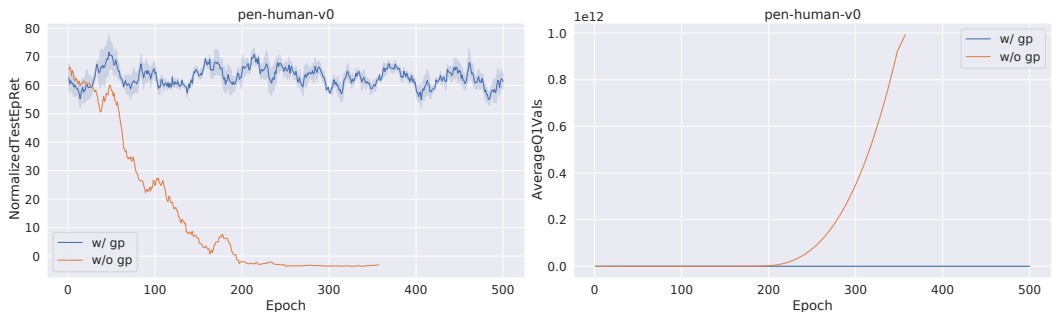

Figure 1: Left: learning curve of pen-human-v0 task. Right: the average Q value of the first ensemble over the course of training. Both curves are smoothed by a factor of 20.

**Performance on datasets with human demonstrations**  The performance on Adroit tasks is shown in Table 2. These tasks are substantially harder than OpenAI gym tasks due to limited training data in a high dimensional observation and action space. Our method makes non-trivial improvement over the behavior cloning. Compared with the state-of-the-art approaches, our approach is superior on half of the tasks and matches the asymptotic performance on the remaining ones. Figure 1 shows that the Q value is bounded when the gradient penalized policy evaluation technique is employed. On the contrary, the Q value without the gradient penalized policy evaluation increases exponentially. Note that we use MMD with Laplacian kernel for Adroit tasks. We observe that the KL-based regularization struggles with datasets collected with narrow behavior distributions (have large density within a tiny space and almost zero density anywhere else). In such a case, the KL divergence is sensitive to tiny policy changes, making gradient-based optimization hard to converge.

## 5.2  ABLATION STUDY

To answer question (2), (3) and (4), we conduct a thorough ablation study on BRAC+ on various tasks with different data collection policies (hopper-mixed, walker2d-medium-expert, halfcheetah-medium).

**Sampled-based MMD vs.  analytical upper bound KL**  The results of using sampled-based MMD versus analytical upper bound KL are shown in Figure 2a. The difference of the performance in the single-modal dataset (halfcheetah-medium) is negligible. However, the performance

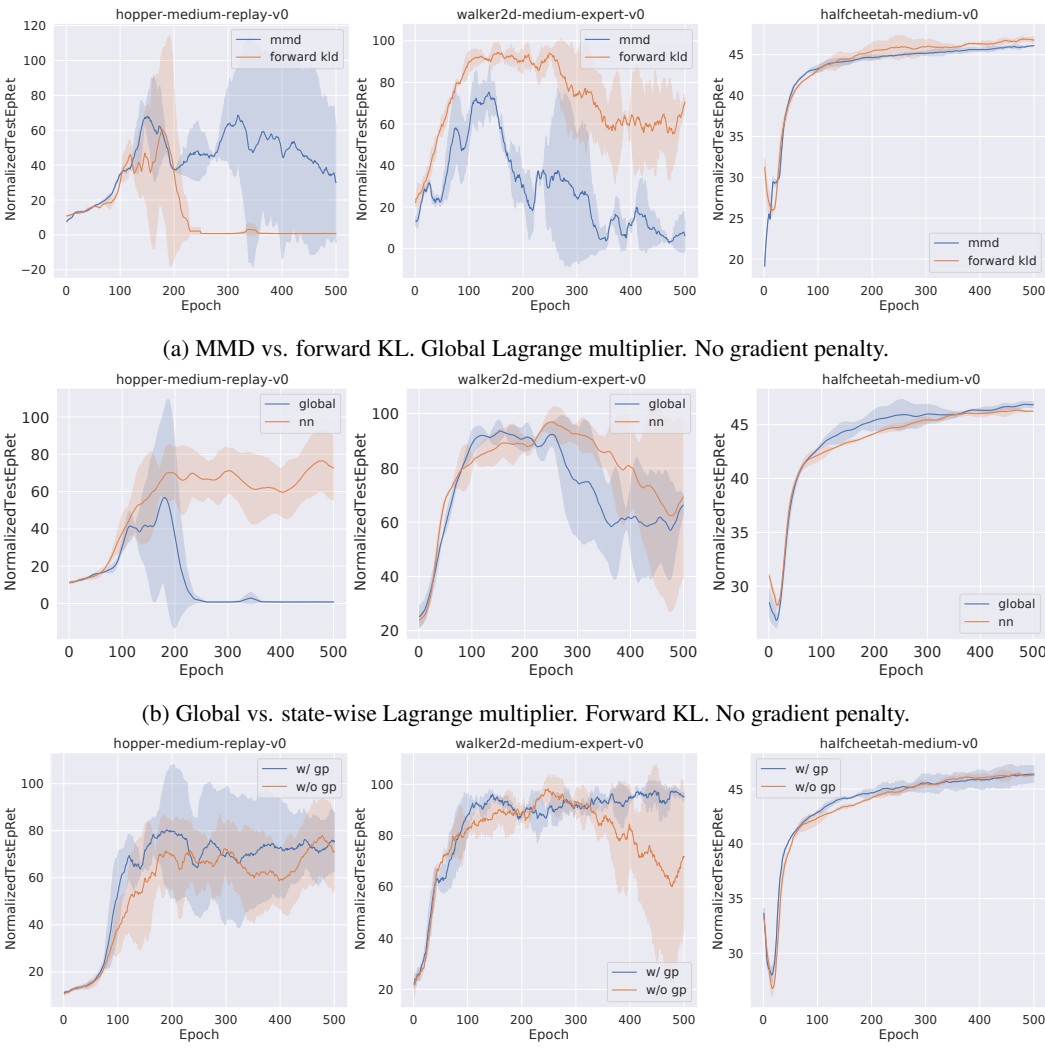

(a) MMD vs. forward KL. Global Lagrange multiplier. No gradient penalty.

(b) Global vs. state-wise Lagrange multiplier. Forward KL. No gradient penalty.

(c) Gradient penalty vs. No gradient penalty. State-wise Lagrange multiplier. Forward KL.

Figure 2: Figures of ablation study. Each setting is repeated for 4 random seeds. The curve is the mean and the shaded area is the standard deviation. The curves are smoothed by a factor of 20. The number of gradient steps per epoch is 2000. To make fair comparison, we only substitute KL divergence with MMD-based measurement with additional MMD-specific hyperparameter tuning. The other design choices are different from (Kumar et al., 2019). Details can be found in Appendix D.

on multi-modal datasets is varied. Our toy example in Appendix B suggests that MMD and the backward KL divergence tends to cover all the "modes" in the behavior policy while the forward KL divergence tends to seek one of the "mode" in the behavior policy. Note that this argument is only valid if the learned policy is single-modal (e.g. Gaussian distribution). The superior performance in the walker2d-medium-expert when using forward KL regularization supports this argument. The results in hopper-mixed task seems to be contradictory. We hypothesize that since the analytical KL has low variance, the policy quickly adopts the out-of-distribution actions when available while in sample-based methods, such adoption is slower due to higher variance in the policy regularization.

**State-wise vs. global regularization** The performance of using global and state-wise regularization on various tasks is shown in Figure 2b. It is noticeable that the state-wise regularization improves the performance in hopper-mixed and walker2d-medium-expert task, while the performance in halfcheetah-medium does not show much improvement. To understand the consequence of the state-wise regularization, we plot the histogram of the KL divergence of all the states in the dataset

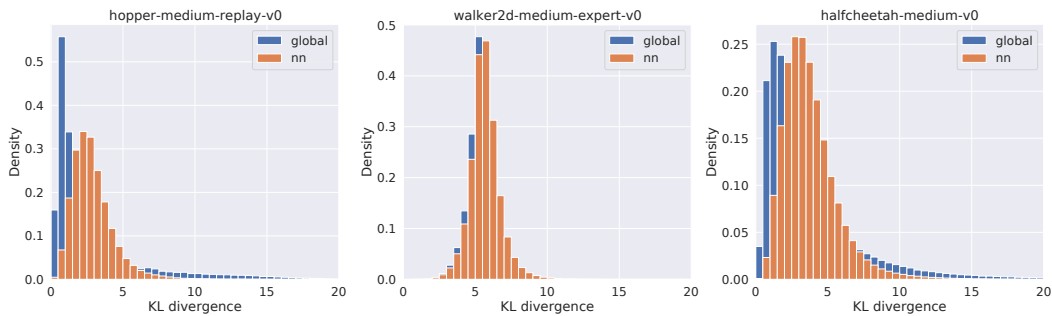

Figure 3: Histogram of the KL divergence of all the states in the dataset with global and state-wise Lagrange multiplier.

in Figure 3. The KL divergence histogram of using state-wise regularization is more concentrated around the threshold while the global regularization is more sporadic. Strictly enforcing the policy regularization around the threshold helps avoid out-of-distribution actions, that is often caused by a few states in the dataset.

**Gradient penalty vs. no gradient penalty** Even with policy regularization, evaluating the Q value at the out-of-distribution actions can't be fully avoided. Thus, it is important to bound the Q value at the out-of-distribution actions such that their values are not greater than the Q value of in-distribution actions. Figure 2c shows the performance with and without gradient penalty in the policy evaluation step. While there is little difference in the hopper-mixed and the halfcheetah-medium task, the performance in the walker2d-medium-expert task stabilizes with the gradient penalized policy evaluation. On the contrary, the performance deteriorates over time without gradient penalty. Please refer to Appendix D for more analysis.

## 6 DISCUSSIONS AND LIMITATIONS

There are several limitations of our approach. The discussion in Appendix D suggests that although KL-regularized offline policy optimization is good at combining sub-optimal policies, they may stuck at local optimums; and they are hard to escape. Another drawback is that the threshold value is hard to set. In our experiments, we try a few numbers that are above the minimum possible KL threshold, which is obtained by training a policy that minimizes that KL divergence.

Finally, we conjecture that the behavior-regularized approach is not sufficient to tackle offline RL problems since it fully ignores the state distribution. To see this, we can create a dataset that only adds a few trajectories from an expert policy to a dataset collected by a low-quality policy. If the low-quality policy doesn't visit the "good" states in the expert policy (can't combine sub-optimal policies), behavior-regularized approach leads to a policy that imitates the expert policy. Such imitation is likely to fail due to compounding errors (Ross et al., 2010). The right approach for this dataset is to completely ignore expert trajectories and combine sub-optimal policies in the low-quality regions. To achieve this, we need to consider the state distribution as the density of the "good" states is very low.

## 7 CONCLUSION

In this paper, we improved the behavior regularized offline reinforcement learning by proposing a low-variance upper bound of the KL divergence estimator to reduce variance, state-dependent Lagrange multiplier to allow more freedom of deviation to high probability states while restricting low probability states and gradient penalized policy evaluation such that the Q values of out-of-distribution actions are not greater than those of in-distribution actions. Our experimental results on challenging benchmarks illustrate the benefits of our improvements.

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

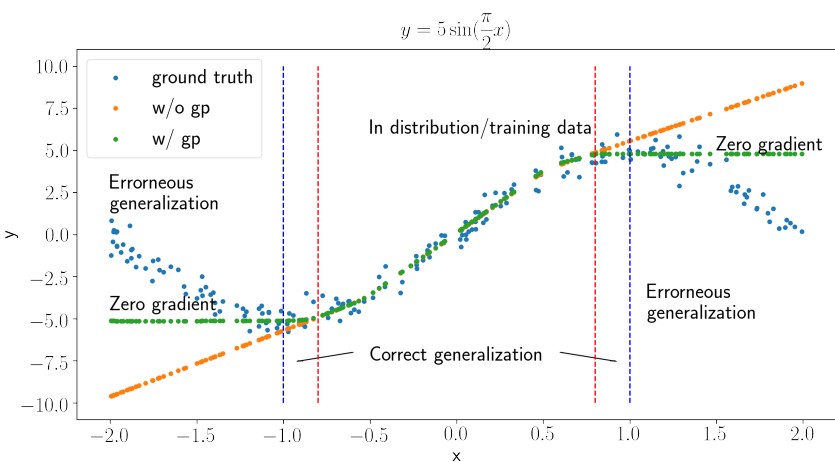

Figure 4: Fitting a regression model with out of distribution data

## A   TOY EXAMPLE TO DEMONSTRATE BOUNDING THE VALUE OF THE OUT OF DISTRIBUTION INPUTS VIA GRADIENT PENALTY

To demonstrate the effectiveness of the gradient penalty to enforce the bound of the predicted values of a regression model, we conduct experiments by fitting a regression model with in-distribution data while minimizing the norm of the gradient at out of distribution inputs. Specifically, we generate dataset $\{(x_i, y_i)\}_{i=1}^{N=100}$ as $x_i \sim U(-0.8, 0.8)$, $y_i = 5\sin(\frac{\pi}{2}x_i) + \epsilon_i$, where $\epsilon_i \sim \mathcal{N}(0, 0.5)$. In addition, we generate out of distribution dataset $\{\tilde{x}\}_{j=1}^{M=100}$ as $\tilde{x} \sim U(-2, -0.8) \cup U(0.8, 2)$. We fit a regression model $f$, represented as a two-layer feed-forward neural network. The size of the hidden layer is 64 and the activation is RELU. We use Adam (Kingma & Ba, 2015) optimizer with learning rate 0.01. In addition to the standard MSE loss, we add a gradient penalty term inspired from (Gulrajani et al., 2017) such that the gradient at the out of distribution inputs is penalized. The overall loss function is:

$$\mathcal{L} = \frac{1}{N}\sum_{i=1}^{N}(f(x_i) - y_i)^2 + \lambda \cdot \frac{1}{M}\sum_{j=1}^{M}||\nabla_{\tilde{x}}f(\tilde{x}_j)||_2 \tag{11}$$

In our experiments, we set $\lambda = 0.1$. We fit $f$ with and without the gradient penalty term. Figure 4 shows the predicted value at both in-distribution and out-of-distribution inputs. The results suggest that: 1) both models generate good prediction for in-distribution inputs ($[-0.8, 0.8]$), which is between the two red lines 2) both models generalize well at the out-of-distribution regions between the red line and the blue line ($[-1, -0.8]$ and $[0.8, 1]$). 3) both models erroneously generalize beyond the blue lines ($[-2, -1]$ and $[1, 2]$). However, the value of the model trained without gradient penalty keep on increasing or decreasing while the model trained with gradient penalty has zero gradient. This suggests that with proper out-of-distribution regularization, the gradient direction at the out-of-distribution inputs point to the in-distribution regions.

## B   VISUALIZATION OF VARIOUS PROBABILITY DIVERGENCE MEASUREMENT

To understand the impact of forward KL ($\mathcal{D}_{\text{KL}}(\pi_\theta || \pi_b)$), backward KL ($\mathcal{D}_{\text{KL}}(\pi_b || \pi_\theta)$) and MMD distance in constraining the learned policy within the support set defined by the behavior policy, we create a toy example as shown in Figure 5. The black curves in the two figures represent the behavior distribution $\pi_b$. The orange, blue and green curves in the graph show forward KL, backward KL and MMD with Laplacian kernel between $\pi_b$ and $\pi_\theta = \mathcal{N}(x, \sigma)$, respectively, where $x$ is a variable between $[-10, 10]$ and $\sigma$ is fixed. If we draw a horizontal line (red), the area below this line defines the support set. In the left figure, the behavior policy is single-modal and there is little difference between the three metrics. However, when the behavior policy is multi-modal, the forward KL divergence contains two bottoms, each corresponds to the peak probability density of

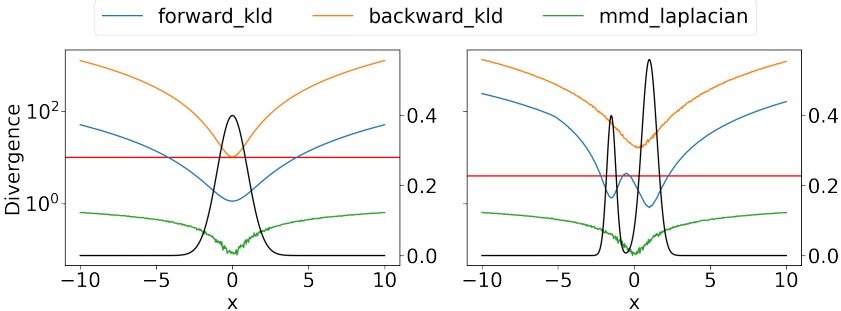

Figure 5: The black curve in the two figures represents the behavior distribution $\pi_b$. The orange, blue and green curves in the graph show forward KL, backward KL and MMD with Laplacian kernel between $\pi_b$ and $\pi_\theta = \mathcal{N}(x, \sigma)$, respectively, where $x$ is a variable between $[-10, 10]$ and $\sigma$ is fixed.

the behavior policy. However, in backward KL and MMD distance, the policy with the smallest divergence actually has low probability density in the behavior policy. This is because minimizing the forward KL leads to *"mode seeking"* while minimizing the backward KL and MMD leads to *"mode covering"*. These two terms are heavily used when talking about generative models such as VAEs and GANs (Ke et al., 2019). When performing offline RL in multi-modal dataset, it is necessary to combine sub-optimal actions in the behavior policy, which is in fact *"mode seeking"*. Mode covering typically leads to out-of-distribution actions even when the divergence measurement is small (e.g. the area between the two modes are actually out-of-distribution) while mode seeking may be stuck at a local optimum (e.g. moving across different "modes" is hard).

In the left figure, $\pi_b = \mathcal{N}(0, 1)$. In the right figure, $\pi_b$ is a mixture of two Gaussian distributions: $\mathcal{N}(-1.5, 0.3)$ and $\mathcal{N}(1, 0.5)$. The weight of each component is 0.3 and 0.7, respectively. $\sigma$ is set to 0.2.

## C  MISSING BACKGROUND

### C.1  VARIATIONAL AUTO-ENCODER

A variational auto-encoder (VAE) (Kingma & Welling, 2014) is a generative model that aims to learn the data distribution $p(X)$ given a set of observations $\{x_i\}_{i=1}^N$. While directly optimizing $p(X)$ is intractable, we can optimize its evidence lower-bound (ELBO):

$$\log p(X) \geq \mathbb{E}_{z \sim q(z)}[\log p(X|z)] - \mathcal{D}_{\text{KL}}(q(z)||p(z)) \tag{12}$$

where $q(Z)$ is the variational distribution and $p(Z)$ is the prior. In VAE, $q(Z)$ is $q(Z|X)$ so that it is an auto-encoder. Optimizing Equation 12 using gradient descent requires back-propagate the gradient through a sample operator. Fortunately, if the latent variable is a multivariate Gaussian distribution, we can use re-parametrization trick. The tightness of the upper bound is the KL divergence between the approximated posterior distribution and the true posterior distribution.

## D  MISSING ABLATION STUDY

For fair comparison between MMD and KL divergence in policy regularization, we only substitute KL divergence with MMD-based measurement with additional MMD-specific hyperparameter tuning. In the original implementation (Kumar et al., 2019), the author uses 4 ensembles of Q network and compute the Q value by a convex combination of both the minimum of the ensembles and the maximum of the ensembles. In our implementation, we follow the standard architecture in the online setting: we only use 2 ensembles of Q networks and compute the Q value as their minimum. In this work, we only compare against MMD with Laplacian kernels. The MMD-specific hyper-parameters are shown in Table 3.

**Difference in training the behavior policy** (Kumar et al., 2019) learns the behavior policy as a $\beta$-VAE (Higgins et al., 2017) with MSE reconstruction loss. This is equivalent to maximizing the log probability of a Gaussian distribution with fixed variance as the decoder output. However, the variational lower bound does not hold in $\beta$-VAE that breaks our derivation. In this paper, we learn the behavior policy as a regular VAE. The decoder outputs a Gaussian distribution with input-conditioned mean and variance.

**Gradient penalized policy evaluation** Figure 6 suggests that the L2 norm of the gradient at the out-of-distribution actions with the gradient penalty is much lower than that without the gradient penalty.

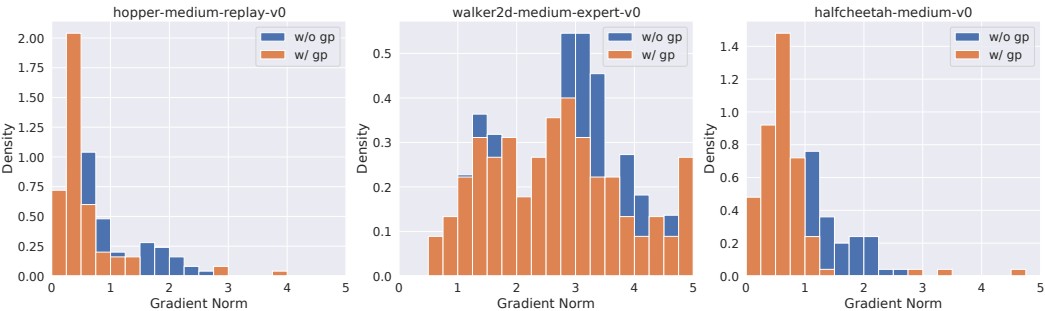

Figure 6: Histogram of the L2 norm of the gradients for out-of-distribution actions

**Out-of-distribution generalization** Lastly, we discuss the out-of-distribution generalization. In offline RL, we rate the learned policy with three levels. Level I policies are able to strictly follow the behavior policy. Level II policies are able to combine sub-optimal policies in the bahevior policy. Level III policies are able to generalize to out-of-distribution actions. To visualize the out-of-distribution generalization, we plot the KL divergence of the learned policy on the testing state distribution in Figure 7. In the hopper-mixed and halfcheetah-medium task, there are periodic spikes, suggesting that the agents visit states which are not in the training distribution at test time. Although the results suggest that such generalization is correct, our approach is unable to explicitly quantify it. Thus, we leave it as future work.

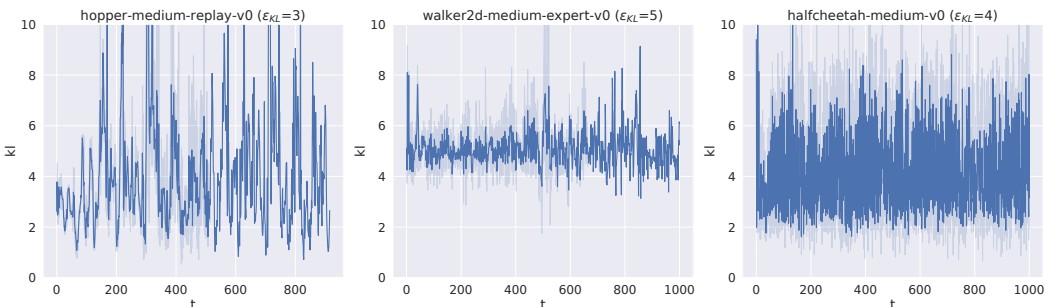

Figure 7: The KL divergence between the learned policy and the behavior policy on the testing data distribution. The $x$ axis is the step in an episode.

**Performance sensitivity to the choices of target entropy** During our experiments, we found that the performance using the KL divergence as the behavior regularization method is very sensitive to the choice of the target entropy. Policies trained with a larger entropy may not be able to capture the narrow expert distribution within the behavior policy distribution. This phenomenon is most typical in the halfcheetah-medium-expert dataset as depicted in Figure 8.

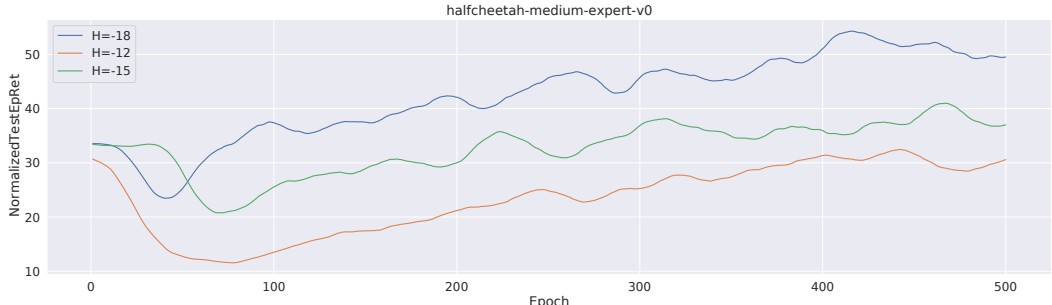

Figure 8: The performance of the halfcheetah-medium-expert task on various choices of the target entropy.

Table 3: MMD-specific Hyper-parameters

| Hyper-parameter | hopper-medium-replay | walker-medium-expert | halfcheetah-medium |
|---|---|---|---|
| $\sigma$ in Laplacian kernel | 10 | 20 | 10 |
| $\epsilon_{\text{MMD}}$ | 0.2 | 0.2 | 0.3 |
| Number of samples | 10 | 10 | 10 |

## E  MISSING RELATED WORK

For KL regularized policy improvement with fixed temperature, the optimal policy has a closed form solution (Wang et al., 2020). Although it avoids estimating the behavior policy, the issue with fixed temperature is that all the states are penalized equally. This leads to over-pessimistic for states with higher occurence in the dataset and over-optimistic for states with lower occurence. (Fox, 2019) presents a closed-form expression for the regularization coefficient that completely eliminates the bias in entropy-regularized value updates. However, the `softmax` operator introduced by the approach makes it hard to use in continuous action space.

## F  IMPLEMENTATION DETAILS

**Computation of the analytical KL upper bound**  To reduce the variance when computing the analytical KL upper bound, we sample $L$ latent variable $z$ and compute the average of the $L$ KL upper bounds. In our experiments, we set $L = 5$. Note that it doesn't reduce the bias of the upper bound.

**Reward scaling**  Any affine transformation of the reward function does not change the optimal policy of the MDP. In our experiments, we rescale the reward to $[0, 1]$ as:

$$r' = (r - r_{min})/(r_{max} - r_{min}) \qquad (13)$$

where $r_{max}$ and $r_{min}$ is the maximum and the minimum reward in the dataset.

**Entropy regularization**  The KL divergence is the sum of negative entropy of the learned policy plus the cross entropy between the learned policy and the behavior policy: $\mathcal{D}_{\text{KL}}(\pi_\theta(\cdot|s), \pi_\beta(\cdot|s)) = -\mathcal{H}(\pi_\theta(\cdot|s)) + \mathcal{H}(\pi_\theta(\cdot|s), \pi_\beta(\cdot|s))$. When the learned policy distribution violates the KL constraints, the KL divergence between the policy distribution and the behavior distribution is decreased by the optimizer. This is equivalent to increasing the entropy of the learned policy and decreasing the cross entropy between the learned policy and the behavior policy. In soft actor-critic (Haarnoja et al., 2018a), the minimum entropy of the learned policy is enforced to encourage exploration. However, due to the absence of exploration, stochastic policy with large entropy will sample out-of-distribution actions when computing the target Q values in Equation 2. If such values are overestimated, the policy will exploit the erroneous Q values when performing the policy improvement in Equation 8 and lead to failure, which can't be corrected without more data. Thus, we maintain the maximum entropy of the learned policy using the technique proposed in (Haarnoja et al., 2018b).

**Initialization**  If the dataset is collected using a narrow policy distribution in a high dimensional space (e.g. human demonstration), the constrained optimization problem using dual gradient descent finds it difficult to converge if random initialization is used for the policy network. To mitigate this issue, we start with a policy that has the minimum KL divergence with the behavior policy: $\pi_\theta = \arg\min_{\pi_\theta \in \Pi} \mathcal{D}_{\mathrm{KL}}(\pi, \pi_\beta)$, where $\Pi$ represents a family of policy types. In this work, we consider $\Pi$ as Gaussian policies. Correspondingly, we initialize the Q network to $Q^{\pi_\theta}$.

**Policy network**  Our policy network is a 3-layer feed-forward neural network. The size of each hidden layer is 512. We apply RELU activation (Agarap, 2018) after each hidden layer. Following (Haarnoja et al., 2018a), the output is a Gaussian distribution with diagonal covariance matrix. We apply `tanh` to enforce the action bounds. The log-likelihood after applying the `tanh` function has a simple closed form solution. We refer to (Haarnoja et al., 2018a) Appendix C for more details.

**Q network**  Following (Haarnoja et al., 2018a; Fujimoto et al., 2018a; Wu et al., 2019), we train two independent Q network $\{Q_{\psi_1}, Q_{\psi_2}\}$ to penalize uncertainty over the future states. We maintain a target Q network $\{Q_{\psi'_1}, Q_{\psi'_2}\}$ with the same architecture and update the target weights using a weighted sum of the current Q network and the target Q network. When computing the target Q values, we simply take the minimum value of the two Q networks:

$$Q_{\psi'}(s', a') = \min_{j=1,2} Q_{\psi'_j}(s', a') \tag{14}$$

Each Q network is a 3-layer feed-forward neural network. The size of each hidden layer is 256. We apply RELU activation (Agarap, 2018) after each hidden layer.

**Behavior policy network**  Following the previous work (Fujimoto et al., 2018b; Kumar et al., 2019), we learn a conditional variational auto-encoder (Kingma & Welling, 2014) as our behavior policy network. The encoder takes a pair of states and actions, and outputs a Gaussian latent variable $Z$. The decoder takes sampled latent code $z$ and states, and outputs a mixture of Gaussian distributions. Both the architecture of the encoder and the decoder is a 3-layer feed-forward neural network. The size of each hidden layer is 512. The activation is relu (Agarap, 2018). To avoid epistemic uncertainty, we train $B$ ensembles of behavior policy networks. At test time, we randomly select one model to perform the calculations. We found $B = 3$ is sufficient for all the experiments. We pre-train the the behavior policy network for 400k gradient steps.

**$\alpha$ network**  The $\alpha$ network takes in a state and outputs the Lagrange multiplier for the state. The architecture of the $\alpha$ network is a 3-layer feed-forward neural network with relu (Agarap, 2018) activation. The size of each hidden layer is 256. We use `softplus` activation after the output to ensure that all the values are positive.

Table 4: Default hyper-parameters

| Hyper-parameter | Value (Gym/Adroit) |
|---|---|
| Optimizer | Adam (Kingma & Ba, 2015) |
| Policy learning rate | 5e-6/5e-8 |
| Q network learning rate | 3e-4 |
| $\alpha$ learning rate | 1e-5/1e-7 |
| batch size | 100 |
| Target update rate $\tau$ | 1e-3 |
| Discount factor $\gamma$ | 0.99 |
| Initial $\beta$ | 10 |
| $\beta$ learning rate | 1e-3 |
| Steps per epoch $T$ | 2000 |
| Number of epochs | 500 |

Table 5: Task-specific hyper-parameters for OpenAI gym tasks.

| Task name | KL divergence threshold $\epsilon_{\text{KL}}$ | Maximum entropy $\mathcal{H}_0$ |
|---|---|---|
| halfcheetah-rand | 9 | -3 |
| walker2d-rand | 0.1 | 6 |
| hopper-rand | 3 | -3 |
| halfcheetah-med | 4 | -12 |
| walker2d-med | 2.1 | -9 |
| hopper-med | 2.4 | -6 |
| halfcheetah-med-exp | 11.5 | -24 |
| walker2d-med-exp | 5 | -12 |
| hopper-med-exp | 2.6 | -6 |
| halfcheetah-mixed | 6 | -12 |
| walker2d-mixed | 4 | -6 |
| hopper-mixed | 3 | -3 |

Table 6: Task-specific hyper-parameters for Adroit tasks with human demonstrations.

| Task name | MMD threshold $\epsilon_{\text{MMD}}$ | Minimum entropy $\mathcal{H}_0$ |
|---|---|---|
| pen-human | 0.06 | -200 |
| hammer-human | 0.1 | -60 |
| door-human | 0.1 | -60 |
| relocate-human | 0.1 | -60 |

