# OpenReview forum: "BRAC+: Going Deeper with Behavior Regularized Offline Reinforcement Learning"
_ICLR.cc/2021/Conference — Reject_

### Official Review · AnonReviewer1 · 2020-10-25

**Rating:** 5
**Confidence:** 4

**Review:**

The paper proposes a number of improvements to the standard behavior-regularized offline RL paradigm introduced in BRAC (Wu 2019). Specifically, the paper proposes (1) using an upper bound on the KL regularization to induce lower-variance updates, (2) state-dependent Lagrange multipliers for the regularization, (3) adaptive causal entropy regularization, (4) behavior-cloned policy initialization, and (5) policy evaluation masking. The paper combines these improvements to propose BRAC+ and presents empirical demonstrations showing favorable performance.

Strengths:

-- Given the many improvements over BRAC suggested, the paper is organized well, and generally easy to follow.

-- The derivations used in improvement (1) are novel as far as I can tell. These seem useful, and I am not aware of previous RL work doing this.

-- The reasoning and motivation behind improvement (2) is especially compelling.

Weaknesses:

-- The derivations for improvement (1) suggest that the behavior policy should be given in variational form (i.e., trained as a VAE). However, the "Initialization" section suggests the behavior policy is actually a Gaussian. So how is improvement (1) applied?

-- The motivation behind improvement (1) is for variance reduction. However, if the aim in variance reduction, aren't there other (simpler?) ways to do this? For example, one can change the KL regularization to a different divergence that is analytically expressible. Or, if one is already using a Gaussian distribution for both behavior and learned policy, the KL is already analytically expressible. These and other alternatives should be considered in addition to the proposed method.

-- I am not sure how the motivation for improvement (2) connects to the proposed solution (state-conditioned Lagrange multipliers). In fact, Sec 3.2 suggests that the KL limit (\epsilon_KL) is constant for all states.  Using the reasoning of the paper, shouldn't this limit be larger for states which appear more in the dataset?

-- Improvement (4), specifically initializing the learned policy as the behavior policy, is used by previous work. For example, https://arxiv.org/abs/2006.03647 I also know that it is used in the implementation of CQL (although I am not sure if the CQL paper mentions it).

-- Improvement (5) seems drastic. Why should we remove all learning signal in areas where the KL is too high? Moreover, the underlying problem here appears to be an issue with the form of KL regularization. Perhaps a solution is to use a different regularizer? One which does not blow up like this?

-- The experimental results are favorable for BRAC+, but they are not especially compelling. As the paper mentions, BRAC+ mostly improves on multi-modal datasets, where as performance of BRAC+ is worse than existing methods on other datasets.

-- Moreover, experimentally, if the claim is that BRAC+ is better on multi-modal datasets, it would be nice to show performance on other (more multi-modal) datasets in D4RL. For example, pointmaze, antmaze (diverse), kitchen -- these all exhibit more multi-model datasets than the Gym Mujoco envs.

---

> ### Author Response · Authors · 2020-11-15
> **Comments for your valuable feedback**
>
> 1. The behavior policy is a VAE (The output distribution of the decoder is a Gaussian) and the learned policy is a Gaussian.
> 2. We can’t assume any prior distribution on the behavior policy. This is the reason that we fit it as a VAE. Since the probability distribution of a VAE is implicit (integral for all the latent variables times the output distribution of the decoder), it is in general hard to compute any divergence measurement in an analytical form. If the reviewer has some suggestion on any better divergence term, we will be glad to try that.
> 3. In the original BEAR approach, the constraints are enforced over the mean divergence in a batch. Using state-wise Lagrange term enforces each state to stay close to the threshold (Our ablation study shows it). We agree that a state-wise threshold is actually necessary and the value should be dependent on the state density. See the limitation sections for more discussions.
> 4. We move the initialization to the implementation detail of the Appendix.
> 5. We replace the q-masking with gradient penalized policy evaluation, which is more reasonable and intuitive for the problem identified in the paper. Please ask more questions and we can further discuss it.

---

> > ### Comment · AnonReviewer1 · 2020-11-23
> > **Response**
> >
> > Thank you for the updates and clarifications. I will update my score accordingly. I'm glad you were able to find an alternative to Q-masking.
> >
> > I also appreciate the additional experimental results. Still, I feel that the results of BRAC+/+ are good, but not compelling. It appears there is only a strong advantage on pen-human. While this suggests BRAC+/+ is a reasonable offline RL algorithm, it's unclear if it is worth all the additional machinery.

---

### Official Review · AnonReviewer4 · 2020-10-26
**A useful set of BRAC enchantments that significantly improves performance of the approach.**

**Rating:** 5
**Confidence:** 4

**Review:**

This paper introduces several enhancements of BRAC (Behavior Regularized Actor Critic), an approach for offline reinforcement learning. The original idea of BRAC is to add a divergence term between behavior distribution, which was used to collect data, and training policy. In the original paper, this is implemented by fitting behavior policy with a density model and then using this density model to estimate a divergence between these distribution. This paper proposes to improve this approach in several ways: first, authors propose to increase the expressiveness of the behavior policy density model by using a mixture of Gaussian distributions; second, authors propose to adaptively tune the weight of the KL term by learning a state-dependent Lagrange multiplier; finally, authors add an additional entropy term, supervised pre-training and out-of-distribution action masking for target Q.

In overall, this paper introduces a simple and well motivated set of enhancements.  The paper is well written and easy to follow. The approach is evaluated on a subset of tasks from a widely used benchmark for offline RL, D4RL. The approach is competitive with the state-of-the-art methods and  outperforms the original BRAC and CQL, a recent state-of-the-art approach, on several tasks.

It is very impressive that such a simple set of enhancements provides significant performance gains. I like the paper but I have several concerns regarding the significance of the contribution and some experimental details:
1) although the approach demonstrates impressive results, the novelty is incremental;
2) the approach improves BRAC by moving from a single Gaussian distribution to a mixture of distributions. How sensitive is the approach is the choice of a number of mixture components?
3) table 3 raises some concerns regarding generality of the approach since some of the crucial hyper-parameters are task-specific;
4) each ablation in figure 1 is performed on a single task that raises concerns regarding generality of the conclusions.

At the moment, this is a borderline paper. However, if the aforementioned concerns are properly addressed, I will reconsider my rating. I looking forward to authors' response.

---

> ### Author Response · Authors · 2020-11-15
> **Thank you for your valuable feedback**
>
> 1. We replaced the q-masking with gradient penalized policy evaluation, which we think is more novel. It leads to better performance on both single-modal and multi-modal datasets as suggested by the new ablation study. More results will be reported in the following revision.
> 2. In our later studies, we actually found that using GMM as the decoder distribution in a VAE doesn’t actually improve the performance, but it increases the complexity. BEAR learns the behavior policy as a $\beta$-VAE (with more focus on the reconstruction term than the KL term) with MSE reconstruction loss. Minimizing the MSE reconstruction loss is equivalent to maximizing the log probability of a Gaussian distribution with fixed variance as the decoder output. However, the variational lower bound does not hold in $\beta$-VAE that breaks our derivation. In this paper, we learn the behavior policy as a regular VAE. The decoder outputs a Gaussian distribution with input-conditioned mean and variance.
> 3. It is true that the hyper-parameter is task specific. In MOPO (https://arxiv.org/abs/2005.13239) , the parameter is also task specific. This is actually reasonable for various reasons:
>     - The entropy of the behavior policy matters since the entropy of the learned policy must be smaller than the behavior policy. This leads to task-specific entropy regularization terms.
>     - The modality of the dataset matters. In a single-modal dataset, we typically set a lower threshold than that in a multi-modal daset for the same environment. This is because the density of the behavior policy in a single-modal dataset is typically lower than that in a multi-modal daset. This leads to task-specific KL threshold terms
> 6. We re-did all the ablation studies.

---

### Official Review · AnonReviewer2 · 2020-10-26
**Some useful techniques, but could benefit from more insight**

**Rating:** 7
**Confidence:** 4

**Review:**

**Summary**: This paper proposes a few improvements to BRAC, a framework for offline (or batch) RL approaches. Namely, the authors propose using an analytical bound on the action KL to reduce variance, a state-dependent Lagrange multiplier (output by a neural network), an extra entropy term, and a mask on the Q-value updates for KL-constraint violating actions. The authors demonstrate the effectiveness of these techniques on a subset of the D4RL datasets, where they compare favorably on some tasks. Ablations demonstrate the benefit of each technique on a subset of tasks.

**Strong Points**: The writing is clear, presenting each of the improvements well. For the most part, these improvements are also technically correct, or at least fairly well motivated.

Experiments demonstrate that these techniques generally improve the performance of BRAC on the D4RL datasets. D4RL is a recent standardized dataset for offline RL, and results are reported using 4 random seeds, which is generally considered reasonable. Thus, this experimental support appears to be rigorous. The reported results are also competitive with other model-free and model-based offline RL approaches. In addition, the authors also perform ablation experiments to investigate each of the proposed improvements, generally demonstrating improved stability and final performance.

The approach is likely to be reproducible, considering that BRAC is a fairly straightforward framework and the proposed improvements are not incredibly complex. The authors provide network architectures and training details in the appendix. However, in practice, many RL algorithms are prone to small, important implementation details. Releasing code would help to further improve reproducibility.

Although none of the improvements, on their own, are particularly novel, their combination appears to provide important practical advances for BRAC. These seemingly small improvements can make a significant difference in practice, e.g. SAC is substantially worse without the ensemble of Q-networks (from TD3).

**Weak Points**: As mentioned above, this paper is not substantially novel; it proposes a few practical improvements for an existing framework/algorithm in an existing area (offline RL). While practical improvements are still worthwhile, this does place a greater requirement for either improved empirical performance or new insights into issues with existing algorithms (e.g. the TD3 paper).

Empirical performance improvements are demonstrated, however, most improvements appear to be on the hopper environment. It should also be noted that 3/4 of the ablation experiments are on the hopper environment as well. In other environments, the results (Table 1) are mixed. While the results are encouraging, I do not seem them as being in favor of BRAC+ across the board. Instead, I could see future researchers incorporating the new techniques into other offline RL approaches, e.g. BEAR+, MOPO+, etc. As such, the issues with previous works and the improvements from these techniques should be clearly described and demonstrated.

While the authors do present ablation experiments for their techniques, the reader is left to infer that the performance improvement is due to the reasons given by the authors. This paper would be more convincing and impactful if these issues were shown directly, giving readers insight into the issues with existing methods. For instance, instead of reporting performance from different KL evaluation techniques, the authors could report the empirical variance of each technique, as well as the bias incurred by their bound.

Another weak point of this paper pertains to the experiment details / comparison. The authors appear to have implemented the basic BRAC framework along with their proposed improvements. The results for other methods in Table 1 are from Fu et al., 2020 and other recent papers. However, it’s unclear if these implementations are comparable. For instance, BEAR typically uses a larger ensemble of Q-networks, whereas the authors report using 2 networks. While these results can be useful for benchmarking across the field, they do not necessarily give insights to readers. I would have preferred to see the authors implement their improvements across multiple previously proposed offline RL algorithms (both model-free and model-based) to enable a more direct comparison and evaluation.

**Accept / Reject**: I would lean slightly toward acceptance for this paper. The authors present several techniques for improving offline RL algorithms within the BRAC framework. These practical improvements are often overlooked, but they can have an outsized impact on the community. This is inline with papers like Flow++, TD3, etc. However, as mentioned above, there are aspects of this paper that could be substantially improved. Currently, the paper does not demonstrate clear insight into the issues that are being fixed; this is only shown through improved performance. Likewise, the experiments could be improved to provide a more direct comparison and evaluation of these techniques (see above).

**Questions**:
The TD loss for Q-network training in Algorithm 1 appears to be missing the action KL / entropy at the next step. Was this a purposeful decision?

How much does pre-training help or hurt? This is presented, but performance is not compared in the experiments section.

**Additional Feedback**:

Introduction:
I wouldn’t necessarily say that BRAC was “proposed” in Wu et al., 2019. While they did coin the term BRAC, this is built around previous works, e.g. BCQ, BEAR, etc. This is part of a larger class of relative-entropy policy search algorithms.

Background:
MDP acronym is given twice.

Improving Behavior Regularized Offline Reinforcement Learning:
There’s a connection between the optimal Lagrange multiplier (or “temperature”) and uncertainty in the Q-value estimate (see Fox, 2019: toward provable unbiased temporal-difference value estimation). This should probably be discussed.
“requires large samples to reduce” —> “requires a large number of samples to reduce”
“variational inference model” —> “latent variable model”
“can be estimated analytically”: but this really only applies in special cases, e.g. Gaussian.
Should explain why the behavior policy is not just one Gaussian.
I found the entropy regularization paragraph difficult to follow.

Related Work:
There’s another related paper by Agarwal et al., 2019 (arXiv:1907.04543).

Experiments:
“regularizor” —> “regularizer”
Unclear what is interesting/new in the performance vs. KL threshold section.

---

> ### Author Response · Authors · 2020-11-15
> **Comments for your valuable feedback**
>
> 1. Our code will be published on github once the decision comes out.
> 2. To increase the novelty, we propose to use gradient penalized policy evaluation to avoid OOD actions. We also include more discussions and insights of the existing approach by comparing KL vs. MMD regularization. In addition, we comment on the behavior-regularized approach itself.
> 3. Our finished experiments using the newly proposed technique on single-modal datasets show competitive results against CQL. We will update the paper as more results come out.
> 4. Due to space limitation, we will include a variance of various regularization methods in the Appendix in the following revision. Theoretically, the bias is the KL divergence between the true posterior distribution and the variational distribution. We are unsure how to compute the bias of the upper bound since the true posterior distribution is unknown.
> 5. We added a comparison between MMD with Laplacian kernel and KL-based regularization. To make fair comparison, we only substitute KL with MMD with additional MMD-specific hyper-parameter tuning. The details are in Appendix D. We don’t know how many gradient steps of the BEAR algorithm needs to be run to achieve the performance reported in D4RL paper. For example, if our approach stops at 300 gradient steps (150 epoch), the performance of our implementation of BEAR (differences are summarized in Appendix D) outperforms the performance reported in the D4RL benchmark in all the three tasks in the ablation study. But in general, we observe that the performance by running the BEAR algorithm deteriorates over time. Such an observation is also reported in the CQL paper.
> 6. The action/KL regularization is not included in the Q update. Including the term leads to a different MDP with reward function: r’(s,a)=r(s,a)-\alpha KL(pi,pi_b). (The derivation can be found in the SAC paper (https://arxiv.org/abs/1801.01290) Appendix B.  Without the term, it is the same MDP with state-wise regularization. We notice that the MMD term is not included in the Q update in the BEAR implementation and thus we follow them.
> 7. We will fix various presentation issues as you suggested in the next revision.

---

### Official Review · AnonReviewer3 · 2020-10-27
**Nice paper, non-trivial and practical improvement over BRAC for offline RL**

**Rating:** 7
**Confidence:** 4

**Review:**

This paper proposes two major improvements over BRAC to address the key challenge of offline RL: out-of-distribution action sampling. The first improvement is to replace the sample-based divergence calculation with an analytical upper-bound estimation. The second improvement is to use a state-dependent weight for divergence regularization, which is learned automatically via dual gradient descent. In addition, several small and practical enhancements are proposed. The method is evaluated on simulation benchmarks and demonstrates performance gains over several state-of-the-art baselines, especially on multi-modal datasets.

The paper addresses an important research direction: offline RL, which is crucial in the field where data collection is difficult and data is scarce. The paper is clearly written. Although the method is improvements of an existing approach (BRAC), the improvements are non trivial, novel, and show promising results.

I have three minor questions and suggestions:
1) Intuitively, I think that KL divergence between policies might not be the best measure to prevent OOD action samples.  The regularization of BRAC(++) is to minimize the KL divergence between the current policy and the behavior policy. The KL divergence only cares about the "closeness" between the policies, and totally ignores the actual density of (s, a) in the dataset. For example, although \pi_\beta(s, a) can have a high probability, which allows the \pi_\theta to sample action a frequently on state s, depending on how the behavior policy is sampled to create the dataset, the state s might not be visited often, and thus has a low density in the dataset.

2) BRAC++ has less improvement on single-modal datasets. However, it is these single-modal datasets that improvements are more desired. In multi-modal datasets, a good policy already exists and is used to collect the dataset. So the use of offline RL in multi-modal datasets is not well motivated despite its better performance. To improve the paper quality, I suggest that more analysis and discussions about why BRAC++ did not work well on single-modal datasets are included.

3) In the ablation study (Figure 1), I suggest that the paper adds more curves from different tasks so that it is clear that the observations and conclusions in ablation study are general across different tasks. If page limit is a concern, it is OK to add additional figures in the supplementary material.

Despite my minor concerns, I still think that this is a strong paper that advances the field of offline RL. Thus, I recommend accepting this paper.

---

> ### Author Response · Authors · 2020-11-15
> **Thank you for your valuable feedback!**
>
> 1. We agree with you that behavior regularization itself is not sufficient to fully avoid OOD actions and state distribution should also be considered. We added a discussion in Section 6. Due to the time limit, this will be left as a future direction.
> 2. With gradient-penalized policy evaluation, the performance of our approach on the single-modal datasets improves as suggested by finished experiments. We will keep updating our paper as more experiments finish.
> 3. We added more ablation study figures, discussions and insights.

---

### Author Response · Authors · 2020-11-15
**Summary of changes in this revision**

We would like to thank the reviewers for your valuable feedback. We first summarize the changes we made (in blue text) in the revised version and then answer specific questions raised by each reviewer.
1. We moved the initialization and entropy regularization into the implementation details in the Appendix E. We notice that other methods actually implement them without explicitly mentioning them.
2. Strictly speaking, the behavior policy we fit is not a VAE since the input of the encoder is state+action and the output of the decoder is action (This follows BCQ, BEAR and BRAC). But the Evidence Lower Bound (ELBO) still holds here. For now, we call it VAE for simplicity.
3. The q-masking technique is specifically designed to achieve good performance for the hopper-mixed task. However, it leads to poor performance on single-modal datasets. We replaced it with “gradient penalized policy evaluation”, which we think is more reasonable, intuitive and novel. We illustrate this approach by creating a toy example in the appendix.
4. We focused our ablation study on the three improvements we proposed (analytical KLD vs MMD, state-wise regularization and gradient penalized policy evaluation). We ran three different tasks with different robotics and different dataset collection policies. We also plot more figures to help the explanation.
5. We discussed KL vs. MMD in the ablation study and in appendix A.
6. We discussed the limitations of our approach and the limitations of the behavior regularized offline RL.
7. We are re-running all our experiments in the Gym domain. We will update the comparison with baseline approaches after we finish all the experiments. The performance of the finished tasks is shown in the table. We will conduct experiments on non-gym tasks if time permits.

---

### Author Response · Authors · 2020-11-24
**Final version submitted**

Dear reviewers,

We really appreciate your effort on providing valuable feedbacks for our paper. We just uploaded the final version of our rebuttal revision. Thanks everyone!

---

### Decision · Program_Chairs · 2021-01-07
**Final Decision**

**Decision:**

Reject

**Comment:**

The paper presents an interesting perspective on improving offline RL within BRAC framework.
Given the improvements over BRAC, the paper is well organized and easy to understand.

The overall results pique interest in comparison with more recent Offline/Batch RL papers: BRAC, BEAR, CQL.
The results in this paper bring BRAC-family of methods closer to CQL with a number of practical improvements, and could have impact in practice.

However, the reviewers have slight split over the marginal value of additional machinery. There do remain some concerns:
- KL divergence is not the best metric to capture OOD issues between policies.
- The additional machinery in comparison to CQL may be unnecessary, at least in terms of results.
- The method requires many task-specific key hyper-parameters, which limits the generality of the approach.

I would recommend rejection as it stands. The paper needs more careful empirical analysis that explains what methodical improvements are actually required and which ones only provide marginal bumps.  With multiple task-specific hyper-params, it may be tricky for these ideas to realize their potential if not clearly understood.
Further release of sufficiently documented and easy to use implementation, will probably be required for acceptance since the main argument in the paper are number of technical improvements in BRAC.